# Explaining the Release Mechanism of Ritonavir/PVPVA Amorphous Solid Dispersions

**DOI:** 10.3390/pharmaceutics14091904

**Published:** 2022-09-08

**Authors:** Adrian Krummnow, Andreas Danzer, Kristin Voges, Stefanie Dohrn, Samuel O. Kyeremateng, Matthias Degenhardt, Gabriele Sadowski

**Affiliations:** 1Laboratory of Thermodynamics, Department of Biochemical and Chemical Engineering, TU Dortmund University, Emil-Figge-Str. 70, D-44227 Dortmund, Germany; 2AbbVie Deutschland GmbH & Co. KG, Global Pharmaceutical R&D, Knollstraße, D-67061 Ludwigshafen am Rhein, Germany

**Keywords:** phase behavior, glass transition, release, amorphous phase separation, liquid–liquid phase separation

## Abstract

In amorphous solid dispersions (ASDs), an active pharmaceutical ingredient (API) is dissolved on a molecular level in a polymeric matrix. The API is expected to be released from the ASD upon dissolution in aqueous media. However, a series of earlier works observed a drastic collapse of the API release for ASDs with high drug loads (DLs) compared to those with low DLs. This work provides a thermodynamic analysis of the release mechanism of ASDs composed of ritonavir (RIT) and poly(vinylpyrrolidone-co-vinyl acetate) (PVPVA). The observed release behavior is, for the first time, explained based on the quantitative thermodynamic phase diagram predicted by PC-SAFT. Both liquid–liquid phase separation in the dissolution medium, as well as amorphous phase separation in the ASD, could be linked back to the same thermodynamic origin, whereas they had been understood as different phenomena so far in the literature. Furthermore, it is illustrated that upon release, independent of DL, both phenomena occur simultaneously for the investigated system. It could be shown that the non-congruent release of the drug and polymer is observed when amorphous phase separation within the ASD has taken place to some degree prior to dissolution. Nanodroplet formation in the dissolution medium could be explained as the liquid–liquid phase separation, as predicted by PC-SAFT.

## 1. Introduction

Newly developed active pharmaceutical ingredients (APIs) often suffer from a low aqueous solubility leading to low bioavailability. One possibility to overcome the limiting factor of low solubility and slow dissolution rate of the API is the use of amorphous solid dispersions (ASDs). In ASDs, the amorphous form of an API is molecularly embedded in a polymer matrix. ASDs have several advantages compared to crystalline APIs, such as higher dissolution rates in the aqueous medium and sometimes also enhanced API solubilities in the dissolution medium. Additionally, API supersaturation in an aqueous medium can be maintained over prolonged periods of time due to crystal-growth inhibition by the polymer, enabling more efficient utilization of the administered tablets [1]. To reduce the pill burden for the patient, ASDs with high drug loads (DLs) are often desirable.

However, several publications have revealed the phenomenon of a collapsing drug release rate with increasing DLs [2,3,4,5,6,7]. A remarkable example was observed for RIT/PVPVA ASDs by Indulkar et al. [2]. The authors investigated the release mechanism of ritonavir (RIT)/poly(vinylpyrrolidone-co-vinyl acetate) (PVPVA) ASDs empirically by systematic DL variation and observed a drastic collapse in the drug release rate at 25 wt% DL. Fluorescence microscopy images revealed undissolved RIT-rich domains on the ASD surface. The authors hypothesized that the collapsing RIT release could be attributed to an amorphous phase separation within the dissolving ASD. They also assumed that for RIT/PVPVA ASDs with DL less than 25 wt%, the kinetics of amorphous phase separation within the ASD was much slower than the kinetics of its dissolution into the medium. Thus, both API and polymer appear to dissolve at the same rate, also referred to as congruent release. Based on their observations for ASDs with higher DLs, they proposed that the kinetics of amorphous phase separation within the ASD was faster than the dissolution kinetics leading to a non-dissolving RIT-rich phase at the ASD surface while PVPVA preferentially went into solution.

In another descriptive study, Shi et al. [8] used X-ray fluorescence imaging to determine the onset of water-induced amorphous phase separation in RIT/PVPVA ASDs of various DLs. They concluded that the kinetics of amorphous phase separation at low DLs are too slow to influence the dissolution kinetics, while at higher DLs, the two kinetic effects are competing on the same time scale, and therefore amorphous phase separation negatively affects the RIT release.

In addition to amorphous phase separation within the dissolving ASDs, liquid–liquid phase separation in the aqueous medium surrounding the dissolving ASDs has also been reported in the literature. Indulkar et al. [2] observed RIT-rich nanodroplets in the dissolution medium via dynamic light scattering for RIT/PVPVA ASDs of low DLs. The formation of drug-rich nanodroplets is expected to be advantageous because they can act as a reservoir to maintain the solubility of the amorphous drug in the dissolution medium [9,10,11]. Sun et al. [12] optically monitored the coalescence kinetics of RIT-rich nanodroplets. They observed that temperature and buffer composition had little influence, whereas the initial droplet number influenced the coalescence kinetics to a large extent. PVPVA pre-dissolved in buffer, on the other hand, had a significant inhibiting effect on coalescence compared to the polymer-free system.

Whereas the effects of collapsing RIT release at high ASD DLs have been generally observed and qualitatively discussed in the literature, a sound scientific explanation is still missing. This work, therefore, focuses on a theoretically profound explanation of the release mechanisms using thermodynamic phase diagrams. Ritonavir was chosen as the model drug due to its low crystallization tendency [13], and PVPVA was selected as the model polymer due to its wide application for pharmaceutical ASDs [14]. Water was considered the dissolution medium. Thus, the phase behavior of the binary subsystems (RIT/PVPVA, PVPVA/water, RIT/water), as well as of the ternary system (RIT/PVPVA/water), were investigated. Phase equilibria and glass-transition temperatures were modeled using the Perturbed-Chain Statistical Associating Fluid Theory (PC-SAFT) and the Kwei equation, respectively. To validate the miscibility gap as predicted by PC-SAFT, a novel approach was developed, which enabled reliable determination of the compositions of the two coexisting phases based on glass-transition temperature measurements. Dissolution experiments were conducted for various DLs, and the release profiles were analyzed and explained based on the developed ternary phase diagram.

## 2. Modeling

### 2.1. Solid–Liquid Equilibrium

An equilibrium between a pure solid (crystalline) phase and a liquid (amorphous) phase was considered for calculating the mole fraction solubility $x_{i}^{L}$ of a crystalline API i in a solvent (here water) or a polymer (here PVPVA) and mixtures thereof according to Equation (1):(1)$$x_{i}^{L}=\frac{1}{\gamma_{i}^{L}}\exp\left[-\frac{\Delta h_{i}^{\mathrm{SL}}}{RT}\left(1-\frac{T}{T_{i}^{\mathrm{SL}}}\right)-\frac{\Delta c_{p,i}^{\mathrm{SL}}}{R}\left(\ln\left(\frac{T_{i}^{\mathrm{SL}}}{T}\right)-\frac{T_{i}^{\mathrm{SL}}}{T}+1\right)\right]$$
where T is the temperature and R is the universal gas constant. The melting properties of the component i, namely the melting temperature $T_{i}^{\mathrm{SL}}$, the melting enthalpy $\Delta h_{i}^{\mathrm{SL}}$, and the difference between solid and liquid heat capacities $\Delta c_{p,i}^{\mathrm{SL}}$ of RIT were taken from literature and are shown in Table 1. The solid phase contained pure RIT form II. The liquid phase contained the dissolved RIT and either PVPVA or water or mixtures thereof. The activity coefficient $\gamma_{i}^{L}$ of a component i in the liquid phase accounts for deviations from an ideal mixture and is a function of temperature, pressure, components present, and their mole fractions. All activity coefficients were determined using PC-SAFT.

### 2.2. Vapor–Liquid Equilibrium

Assuming the vapor phase behaves like an ideal gas, the equilibrium between a liquid (amorphous) phase and a vapor phase was calculated by solving Equation (2), with water being the only component present in both the liquid phase and the vapor phase:(2)$$\frac{\mathrm{RH}}{100\%}=\frac{p_{\mathrm{water}}}{p_{\mathrm{water}}^{\mathrm{LV}}}=x_{\mathrm{water}}\;\gamma_{\mathrm{water}}$$
where RH is the relative humidity, $p_{\mathrm{water}}$ and $p_{\mathrm{water}}^{\mathrm{LV}}$ are the partial pressure of water in the vapor phase and the vapor pressure of water, respectively. $x_{\mathrm{water}}$ is the mole fraction of water in the liquid phase containing RIT, PVPVA, and water, and $\gamma_{\mathrm{water}}$ is the activity coefficient of water in that liquid phase obtained from PC-SAFT.

### 2.3. Liquid–Liquid Equilibrium

Liquid–liquid equilibrium calculations were applied to determine the compositions of the two phases of amorphous phase separation (within the ASD) as well as of liquid–liquid phase separation (formation of nanodroplets) in the dissolution medium. The equilibrium compositions of two coexisting liquid (amorphous) phases, L1 and L2, were determined via simultaneously solving Equation (3) for every component i that distributes between the two phases:(3)$$x_{i}^{L1}\gamma_{i}^{L1}=x_{i}^{L2}\gamma_{i}^{L2}$$
where $x_{i}^{L1}$ and $x_{i}^{L2}$ are the mole fractions of component i (being RIT, PVPVA, or water) in phase L1 and phase L2, respectively. When L1 represents the RIT-poor phase, $x_{\mathrm{RIT}}^{L1}$ is the mole fraction solubility of amorphous RIT. All calculated mole fractions were converted into the mass fractions shown in the phase diagrams in this work.

Systems such as RIT/PVPVA/water reveal strong intermolecular interactions, leading to strong deviations from an ideal mixture. The activity coefficients $\gamma_{i}^{L1}$ and $\gamma_{i}^{L2}$ of component i in phase L1 and phase L2 account for these interactions. They depend on temperature, pressure, and the compositions of the corresponding phases and were calculated via PC-SAFT in this work.

### 2.4. Glass Transition

The glass transition temperatures of ternary RIT/PVPVA/water mixtures were calculated via the Kwei equation (Equation (4)) [17]. This is an extension of the Gordon–Taylor equation [18], which assumes an ideal mixture resulting, inter alia, in volume additivity.
(4)$$T_{g}=\frac{\sum_{i}K_{i,\mathrm{PVPVA}}w_{i}T_{g,i}}{\sum_{i}K_{i,\mathrm{PVPVA}}w_{i}}+\sum_{i\neq j}w_{i}w_{j}q_{ij}$$
where $w_{i}$ and $w_{j}$ are the mass fractions of components i and j. The Gordon–Taylor interaction parameters $K_{i,\;\mathrm{PVPVA}}$ between components i and PVPVA were predicted using the Simha–Boyer rule [19], according to Equation (5):(5)$$K_{i,\mathrm{PVPVA}}=\frac{\rho_{\mathrm{PVPVA}}\;T_{g,\mathrm{PVPVA}}}{\rho_{i}T_{g,i}}$$

The glass transition temperature, $T_{g,i}$, of pure RIT and pure PVPVA were measured in this work. The glass transition temperature of water as well as the density, $\rho_{i}$, of the amorphous component, i, were taken from literature. All pure-component densities and glass-transition temperatures used in this work are listed in Table 2.

The Kwei interaction parameters, $q_{ij}$, were adjusted to the glass transition temperature measurements of single-phase binary mixtures measured in this work. The average relative deviation (ARD) was calculated via Equation (6), where $T_{g,\mathrm{cal},k}$ and $T_{g,\exp,k}$ are the calculated and experimental glass transition temperature, respectively, of data point k out of NP data points. The resulting Kwei interaction parameters and the ARDs are summarized in Table 3.
(6)$$\frac{\mathrm{ARD}}{100\%}=\frac{1}{\mathrm{NP}}\overset{\mathrm{NP}}{\underset{k=1}{\sum }}\left|1-\frac{T_{g,\mathrm{cal},k}}{T_{g,\exp,k}}\right|$$

### 2.5. PC-SAFT

Activity coefficients used in this work were calculated using PC-SAFT [24] from the residual Helmholtz energy $a^{\mathrm{res}}$. The latter is the sum of three different contributions according to Equation (7):(7)$$a^{\mathrm{res}}=a^{\mathrm{hc}}+a^{\mathrm{disp}}+a^{\mathrm{assoc}}$$

The hard-chain contribution $a^{\mathrm{hc}}$ considers repulsive interactions between molecules, which are treated as chains of segments and require the segment number $m_{i}^{\mathrm{seg}}$ and the segment diameter $\sigma_{i}$ of each component i. The dispersion contribution $a^{\mathrm{disp}}$ accounts for Van der Waals attractions between the molecules and is obtained using the dispersion energy parameter $u_{i}k_{B}^{-1}$. The association contribution $a^{\mathrm{assoc}}$ caused by hydrogen bond-forming molecules requires additional pure-component parameters. Namely, these are the number of association sites $N_{i}^{\mathrm{assoc}}$, the association-energy parameter $\varepsilon^{A_{i}B_{i}}k_{B}^{-1}$ and the association volume $\kappa^{A_{i}B_{i}}$. $k_{B}$ is the Boltzmann constant. The pure-component parameters used in this work were taken from the literature and are summarized in Table 4.

The combined rules of Berthelot [27] (Equation (8)) and Lorentz [28] (Equation (9)) were applied to determine the segment diameter and the dispersion energy in mixtures of components i and j:(8)$$\sigma_{ij}=\frac{1}{2}\left(\sigma_{i}+\sigma_{j}\right)$$
(9)$$u_{ij}=\sqrt{u_{i}u_{j}}\;\left(1-k_{ij}\right)$$

The binary interaction parameter $k_{ij}$ corrects for deviations from the geometric mean of the dispersion energies of the pure components and might depend on temperature as expressed in Equation (10):(10)$$k_{ij}=k_{ij,a}T^{2}+k_{ij,m}T+k_{ij,b}$$

The coefficients $k_{ij,a}$, $k_{ij,m}$, and $k_{ij,b}$ are usually determined via fitting to data of binary mixtures. $k_{ij,m}$ and $k_{ij,b}$ for RIT/PVPVA and RIT/water were available from the literature [15]. In this work, we chose a parabolic temperature dependency for the $k_{ij}$ of PVPVA/water. Thus, $k_{ij}$ was fitted to gravimetric water-sorption measurements (vapor–liquid equilibrium) from the literature [29] and cloud-point measurements (liquid–liquid equilibrium) from this work. All interaction parameters used in this work are listed in Table 5.

To calculate the association energy and the association volume in mixtures of components i and j, the combining rules of Wolbach and Sandler [30] (Equations (11) and (12)) were used:(11)$$\varepsilon^{A_{i}B_{j}}=\frac{1}{2}\left(\varepsilon^{A_{i}B_{i}}+\varepsilon^{A_{j}B_{j}}\right)$$
(12)$$\kappa^{A_{i}B_{j}}=\sqrt{\kappa^{A_{i}B_{i}}\kappa^{A_{j}B_{j}}}{\left(\frac{\sqrt{\sigma_{i}\sigma_{j}}}{\frac{1}{2}\left(\sigma_{i}+\sigma_{j}\right)}\right)}^{3}$$

## 3. Materials and Methods

### 3.1. Materials

RIT was obtained from AbbVie Deutschland GmbH & Co. KG (Ludwigshafen, Germany). The copolymer PVPVA (Kollidon^®®^ VA64) with a weight-average molar mass of 65,000 g/mol was purchased from BASF SE (Ludwigshafen, Germany). Hydrochloric acid (10%) was purchased from AppliChem GmbH (Darmstadt, Germany), and potassium acetate (purity ≥ 99%) was purchased from Carl Roth GmbH & Co. KG (Karlsruhe, Germany). Acetone (purity ≥ 99.8%), sodium dihydrogen phosphate (purity ≥ 99%), and disodium hydrogen phosphate dihydrate (purity ≥ 99.5%) were purchased from Merck KGaA (Darmstadt, Germany). Magnesium nitrate hexahydrate (purity ≥ 98%), potassium nitrate (purity ≥ 99%), and potassium sulfate (purity ≥ 99%) were purchased from VWR International GmbH (Darmstadt, Germany). All substances were used without further purification. Water was freshly prepared with a Milli-Q^®®^ Advantage A10 purification system from Merck KGaA (Darmstadt, Germany).

### 3.2. Methods

#### 3.2.1. Ball Milling

To manufacture ASDs via vacuum compression molding, physical mixtures were firstly prepared via ball milling. In total, 0.7 g of RIT and PVPVA were filled into a 10 mL stainless-steel cup of the ball mill Pulverisette 23 by Fritsch GmbH (Idar-Oberstein, Germany) together with a Ø15 mm stainless-steel ball. Milling was performed three times for 3 min at 50 Hz with two breaks of 60 s for heat removal. Afterward, the physical mixtures were dried in a vacuum chamber for at least 24 h before further usage.

#### 3.2.2. Vacuum Compression Molding (VCM)

Next, 0.5 g of dried physical mixtures were filled into the 10 mm × 40 mm VCM bar tool from MeltPrep GmbH (Graz, Austria) to produce ASDs for mDSC measurements. The mixtures were heated for 15 min to a temperature of 150 °C. ASDs with high DLs, such as 35 wt%, 40 wt%, and 100 wt%, were manufactured without vacuum to prevent material loss due to the low melt viscosity. When the vacuum was applied in a heating step, it was also applied during the subsequent cooling step of 10 min at 25 °C. The obtained amorphous ASD bar was manually crushed, transferred into the ball mill described above, and pulverized for 60 s at 20 Hz using the same cup and ball dimensions as before. The powder was then transferred into a vacuum chamber for at least 24 h before further use.

ASDs for visual turbidity inspection were manufactured by filling 0.018 g of a dried physical mixture into the ⌀8 mm VCM disc tool from MeltPrep GmbH (Graz, Austria). Depending on the DL, the mixture was heated to an internal temperature of 140–150 °C within 7–20 min under vacuum. During the subsequent cooling step of 10 min at 25 °C, a vacuum was applied again. After production, the discs were stored immediately at the desired conditions.

#### 3.2.3. Spray Drying

Amorphous particles (diameter 1–25 µm [31]) for dissolution experiments were manufactured using the mini spray dryer B-290 with an inert-loop B-295 from BÜCHI Labortechnik AG (Flawil, Switzerland). For RIT/PVPVA ASDs of 10–40 wt% DL, a total of 4 g of RIT and PVPVA in their respective compositions were dissolved in 200 g acetone. For pure amorphous RIT spray-dried particles, 2 g of RIT were dissolved in 200 g acetone. The feed solutions entered the ⌀0.7 mm two-fluid nozzle with a flow rate of 9 mL/min and were dispersed by nitrogen with a flow rate of 0.536 m^3^/h. The inlet temperature was set at 65 °C, and the flow rate of 35 m^3^/h was used for the aspiration of nitrogen. After spray drying, the particles were dried in a vacuum chamber for at least 48 h.

#### 3.2.4. Cloud-Point Measurements

Specific amounts of dried PVPVA were dissolved in 10 g of water to prepare a series of aqueous solutions of known PVPVA concentrations. The cloud-point temperatures of these solutions were determined at a wavelength of 488 nm via a UV-Vis spectrophotometer Specord 210 Plus by Analytik Jena GmbH (Jena, Germany). The slit was set to 1 nm, and the integration time was 0.1 s. A sealed 10 mm path length cuvette with an internal temperature sensor was selected for analysis. A water-operated heat exchanger with a Peltier element by Analytik Jena GmbH (Jena, Germany) was used for temperature control (accuracy ± 0.1 K). The temperature was stepwise increased by 0.2 K/min in the temperature range of 57–97 °C to determine the cloud-point temperature. During the temperature increase, UV-Vis transmission decreased from 100% toward 0% due to increasing turbidity caused by the evolution of a second liquid phase. The onset of the transmission decrease in the transmission vs. temperature curve was considered the cloud-point temperature.

#### 3.2.5. Storage at Humid Conditions

To measure samples stored at 25 °C at constant humid conditions, saturated salt solutions were prepared with an excess of potassium acetate (23% RH), magnesium nitrate (53% RH), potassium nitrate (94% RH), and potassium sulfate (97% RH) [32]. The storage times of the samples were at least 2 days.

#### 3.2.6. Gravimetric Water-Sorption Measurements

The samples were weighed before and after storage under humid conditions to determine the mass increase (accuracy ± 0.005 mg) resulting from water sorption.

#### 3.2.7. Modulated Differential Scanning Calorimetry (mDSC)

Investigations of glass transitions and melting points of samples were performed using a DSC Q2000 apparatus equipped with an RCS90 cooling system from TA Instruments-Waters LLC (Newcastle, DE, USA). The DSC cell was purged with a nitrogen flow rate of 50 mL/min, and the temperature (accuracy ± 0.1 K) was calibrated with pure indium.

The measurement procedure enabled the investigation of the phases present in the sample and the glass transition simultaneously. The absence of crystals or a second amorphous phase was verified for the fresh ASDs manufactured by VCM and spray drying (Appendix A) by the absence of melting peaks or multiple glass transitions, respectively. Glass-transition measurements were performed using the ASDs produced by VCM. Further, 5–10 mg of dried ASD materials were filled into a TZero Hermetic aluminum pan stored at 23% RH, 53% RH, and 94% RH. Pans with samples stored at 0% RH were closed via a pierced TZero Hermetic aluminum lid to allow during the measurement removal of any water (potentially absorbed through sample preparation). Pans with samples stored at other RHs were closed with a TZero Hermetic aluminum lid to prevent the release of water during measurement.

The measurement procedure consisted of several steps with a minimum temperature $T_{\min}$ and a medium temperature $T_{\mathrm{med}}$ of at least 20 K below and above $T_{g}$ of the ASDs, respectively. In the first step, the samples were quench-cooled to $T_{\min}$ and kept isothermally at $T_{\min}$ for 5 min. The samples were then heated at 5 K/min to $T_{\mathrm{med}}$ and held there for 5 min to erase their thermal history. Afterward, the samples were cooled at 10 K/min to $T_{\min}$ again before reheating at 5 K/min to 150 °C. The linear heating rate was always superposed by a sinusoidal temperature modulation with an amplitude of ±0.796 K and a period of 60 s.

The last heating step was used to detect melting peaks of potentially crystalline domains ($T_{\mathrm{RIT}}^{\mathrm{SL}}$ = 124.97 °C [15]). The glass transitions were determined as inflection points of the reverse heat flow curve of the second heating cycle with the software TA Universal Analysis by TA Instruments-Waters LLC (Newcastle, DE, USA).

To measure the freezing point depression of water due to dissolved PVPVA (leading to the aqueous PVPVA solubility at the measured melting temperature, Appendix A), 10–15 mg of these solutions were filled into a TZero Hermetic aluminum pan which was immediately closed with a TZero Hermetic aluminum lid. In the first step, the sample was equilibrated at 20 °C, then it was cooled to −60 °C and held there for 5 min. Afterward, the sample was heated at either 5 K/min or 0.2 K/min to 20 °C. The linear heating rate was superposed by a sinusoidal temperature modulation with an amplitude of either ±0.796 K or ±0.318 K and a period of 60 s. The obtained melting points were used for extrapolation to a heating rate of 0 K/min [33]. All melting points were determined as onsets of the heat flow curve with the software TA Universal Analysis by TA Instruments-Waters LLC (Newcastle, DE, USA).

#### 3.2.8. Dissolution Experiments

For the dissolution experiments, 682.4 g of water was mixed with 14.13 g of phosphate buffer of pH = 8 (25 °C) and 3.5 g of 0.2 M HCl. The resulting dissolution medium of pH = 7 (25 °C) was immediately transferred into a USP II vessel of the dissolution apparatus AT 7Smart from SOTAX AG (Allschwil, Switzerland). The temperature was set to 25 °C (accuracy ± 0.2 K), and the stirrer speed was set to 150 rpm. Then, 0.1 g of the amorphous material (RIT or ASD or PVPVA) stored in a vacuum chamber prior to use was added to the dissolution medium.

Samples were taken manually with syringes and filtered through polyethylene terephthalate (PET) membranes with a pore size of 0.20 µm before dilution with a known aliquot of the dissolution medium. The volume taken from the vessel was immediately replaced with a new dissolution medium. Overall concentrations of RIT and PVPVA were determined simultaneously via a UV-Vis spectrophotometer Specord 210 Plus by Analytik Jena GmbH (Jena, Germany). For this purpose, calibration curves at wavelengths of both 225 nm and 240 nm were recorded for the two components. Samples were analyzed on the UV-Vis spectrophotometer with a slit of 1 nm and an integration time of 0.1 s in a sealed cuvette with a path length of 10 mm. The temperature of the cuvette holder was 25 °C (accuracy ± 0.02 K). A linear superposition of the component spectra to mixture spectra was verified for mixtures of known composition and subsequently used for spectra evaluation of mixtures of unknown compositions. Each dissolution experiment was performed in duplicates, and the mean values were reported.

## 4. Results and Discussion

To explain the release mechanism of RIT from RIT/PVPVA ASDs, we aimed to predict and validate the ternary phase diagram of RIT/PVPVA/water.

### 4.1. Ternary Phase Diagram of RIT/PVPVA/Water

The ternary phase diagram was predicted using the parameters from Table 2, Table 3, Table 4 and Table 5, which were validated against the binary subsystems RIT/PVPVA, PVPVA/water, and RIT/water (Appendix A). This prediction then serves as a basis to explain the release mechanism of RIT from RIT/PVPVA ASDs.

#### 4.1.1. Amorphous Phase Separation and Liquid–Liquid Phase Separation in the Ternary Phase Diagram

When a dry ASD is exposed to an aqueous dissolution medium, generally, two phenomena happen concurrently: (1) the ASD absorbs the dissolution medium (water) and (2) the ASD dissolves in the dissolution medium. For hydrophobic APIs, this might lead to (1) amorphous phase separation in the wet ASD and (2) liquid–liquid phase separation (nanodroplet formation) in the liquid phase surrounding the ASD. These phenomena will be considered in detail in the following, first using the schematic phase diagram of an API/polymer/water system. Many APIs (e.g., RIT) show a huge miscibility gap with water (Figure 1), which still exists when a polymer, such as PVPVA, is added. Generally, the size of the miscibility gap depends on the kind of polymer, API, and temperature. Every initial composition of API/polymer/water located within this miscibility gap leads to the formation of two liquid/amorphous phases.

Starting from a dry and homogeneous ASD (lower side of the triangle in Figure 1), there are two possible pathways into this miscibility gap. When an initially dry ASD is exposed to a dissolution medium of certain water activity (equals RH when in contact with a vapor phase), water is absorbed along the straight line in Figure 1a) between the point denoting the composition of the dry ASD and the apex depicting pure water. Upon water sorption, and depending on the ASD composition, the resulting water-containing ASD might be located within the miscibility gap (point F in Figure 1a)). In this case, demixing occurs along the corresponding tie line resulting in an API-poor phase L1 and an API-rich phase L2 in the ASD (Figure 1c)). The latter is often named amorphous phase separation in the literature. An mDSC measurement of such a phase-separated ASD would typically result in two glass-transition temperatures corresponding to the two phases.

Again, the same miscibility gap applies when an ASD is surrounded by liquid water as the dissolution medium. The only difference is that water concentration in the dissolution medium is much higher than in the wet ASD (Figure 1b)). When an initially dry ASD is dissolved in pure water as a dissolution medium, API and polymer dissolve along the same straight line as before, connecting pure water and the point denoting the composition of the dry ASD. Depending on the amount of ASD dissolved in water, the composition of the resulting dissolution medium containing both API and polymer might be located in the same miscibility gap as before but at very high water concentrations (point F in Figure 1b)). In this case, demixing occurs along the corresponding tie line again, leading to an API-poor phase L1 and an API-rich phase L2, this time in the dissolution medium (Figure 1d)). This liquid–liquid phase separation can be observed as droplet formation in the dissolution medium. It is worth noting that the API concentration in the API-poor phase corresponds to the solubility of the amorphous API in the aqueous polymer-containing medium.

Considering mass conservation results in the lever rule enables calculating the mass ratio of the API-rich phase L2 to the API-poor phase L1 at equilibrium for a given feed point:(13)$$\frac{m^{L2}}{m^{L1}}=\frac{w_{i}^{F}-w_{i}^{L1}}{w_{i}^{L2}-w_{i}^{F}}$$
where $m^{L1}$ and $m^{L2}$ are the total masses of the API-poor phase L1 and the API-rich phase L2, respectively. $w_{i}^{F}$, $w_{i}^{L1}$, and $w_{i}^{L2}$ are the mass fractions of component i in the feed F, the API-poor phase L1, and API-rich phase L2, respectively.

#### 4.1.2. Prediction of the Ternary Phase Diagram

Figure 2 shows the ternary phase diagram of RIT/PVPVA/water predicted by PC-SAFT using the parameters from Table 4 and Table 5. These parameters were determined via fitting the binary subsystems only and were verified for these binary systems (Appendix A). Using only one binary parameter per binary system, all types of experimental data, i.e., solid solubilities, liquid–liquid miscibility gaps, as well as vapor-liquid equilibria of the binary subsystems, could be modeled in near quantitative agreement with the experimental data.

None of the parameters were fitted to the ternary system. As seen in Figure 2, the predicted miscibility gap for the ternary system RIT/PVPVA/water is huge. The predicted tie lines result in phases very poor in RIT and phases that contain almost only RIT. ASDs with low water content are predicted to be homogeneous (below the miscibility gap) and glassy at 25 °C (green region). Mixtures with compositions in the overlapping area of the miscibility gap and the glassy region are prone to demixing but are kinetically stabilized, which can hinder or delay demixing.

#### 4.1.3. Experimental Validation of the Ternary Phase Diagram

To validate the predicted miscibility gap (Figure 2), RIT/PVPVA ASDs with different DLs and various mass fractions of water were analyzed via turbidity inspection and mDSC. Different water concentrations in the samples were realized via storing these samples for at least 2 days at different RHs. Afterward, the water concentrations were determined gravimetrically.

First, the ASDs were stored at 97% RH and visually inspected. As shown in Figure 3, after 2 days of storage, the turbidity and occurrence of a second amorphous phase were visible for ASDs with 5–40 wt% DL. This validates the existence of the amorphous phase separation predicted by PC-SAFT at 97% RH (Figure 2). In samples with DLs higher than 25 wt% (Figure 3g–i), both phases were macroscopically distinguishable. Our results are in accordance with DSC measurements from Purohit et al. [34], which showed two glass-transition temperatures, depicting two phases, for RIT/PVPVA ASDs with DLs between 10 wt% and 50 wt% after exposure to 97% RH.

Second, for the ASDs stored at 23% and 53% RH for at least 10 days and monitored by DSC, only one glass-transition temperature and no melting peaks were detected (see Appendix A, Appendix A). Thus, these samples did not show amorphous phase separation because they are not located within the miscibility gap, as predicted by PC-SAFT (Figure 2).

DSC scans of RIT/PVPVA ASDs with DLs of at least 10 wt% stored at 94% RH for a minimum of 7 days showed two glass transitions and no melting peak (Appendix A, Appendix A and Table 6). The two glass-transition temperatures indicate a phase split into an RIT-rich phase and an RIT-poor phase, which is again in qualitative agreement with the miscibility gap predicted in Figure 2. The glass-transition temperatures of the RIT-rich and RIT-poor phases corresponded to the glass-transition temperatures of pure RIT and pure PVPVA, respectively, at the same RH. This means that the RIT-rich phase contained almost pure RIT, whereas the RIT-poor phase contained almost pure PVPVA confirming the huge miscibility gap predicted using PC-SAFT.

We used the two measured $T_{g}$s per ASD stored at 94% RH (Table 6) to determine the compositions of two evolving amorphous phases. For that purpose, the deviations between the experimental glass-transition temperatures $T_{g,\exp}^{L1}$ and $T_{g,\exp}^{L2}$ and the glass-transition temperatures $T_{g,\mathrm{cal}}^{L1}$ and $T_{g,\mathrm{cal}}^{L2}$ calculated using the Kwei equation (Equation (4)), and the parameters from Table 2 and Table 3 were minimized simultaneously for the two phases, L1 and L2 (Equation (14)).
(14)$$\underset{m_{i}^{L1}\in R,\;\;\;m_{i}^{L2}\in R}{\min}\;{\left(T_{g,\exp}^{L1}-T_{g,\mathrm{cal}}^{L1}\right)}^{2}+{\left(T_{g,\exp}^{L2}-T_{g,\mathrm{cal}}^{L2}\right)}^{2}$$
(15)$$s.t.\;m_{i}^{L1}+m_{i}^{L2}=m_{i}^{F}$$
(16)$$0\leq m_{i}^{L1}\leq m_{i}^{F}$$
(17)$$0\leq m_{i}^{L2}\leq m_{i}^{F}$$

This optimization was performed with respect to the masses $m_{i}^{L1}$ and $m_{i}^{L2}$ of every component i (polymer, API, and water) in the two phases L1 and L2, considering the constraints given by the mass conservation (Equation (15)) as well as the lower and upper bounds for the masses of the components in phases L1 and L2 (Equations (16) and (17)). Total masses $m_{i}^{F}$ of API, polymer, and water were known from the gravimetric measurements of the dry ASDs and of the ASDs after storage at a fixed RH. The mass fractions $w_{i}^{L1}$ and $w_{i}^{L2}$ of the two phases L1 and L2 evolving from the phase separation upon water sorption were then obtained from the component masses according to:(18)$$w_{i}^{L1}=\frac{m_{i}^{L1}}{\sum^{\;}m_{i}^{L1}}$$
(19)$$w_{i}^{L2}=\frac{m_{i}^{L2}}{\sum^{\;}m_{i}^{L2}}$$

The resulting equilibrium concentrations are shown in Figure 2 and Table 6 for all four samples with varying DLs. Regardless of the DL, all equilibrium concentrations determined by the novel approach are almost identical and are in very good agreement with the predictions using PC-SAFT (see Figure 2). Thus, the results not only qualitatively verify the existence of the miscibility gap but, moreover, show that both the length and the slope of the predicted tie lines are in quantitative agreement with the experimental data. To the best of our knowledge, this is the first time ever that a quantitative phase diagram for the system RIT/PVPVA/water is provided. It has been predicted via PC-SAFT solely based on pure-component parameters and experimental data of the binary subsystems and was experimentally validated against various types of experimental data in this work. Based on the same modeling, Figure 4 shows a plot of the predicted phase mass ratios $m^{L2}/m^{L1}$ calculated (Equation (13)) by PC-SAFT, which are in excellent agreement with the experimental data (Figure 4). It is worth mentioning that Figure 4 illustrates the lever rule: the smaller the DL, the smaller the total amount of evolving RIT-rich phase L2. Figure 2 and Figure 4 will be subsequently used to understand and explain the release behavior of RIT/PVPVA ASDs in water.

### 4.2. Dissolution Experiments

#### 4.2.1. Dissolution of RIT/PVPVA ASDs

Figure 5 shows the measured release profiles of (a) PVPVA and (b) RIT from RIT/PVPVA ASDs during the dissolution experiments. While at 10 wt% DL, RIT is completely released, its release decreases at 20 wt% DL and breaks down for all DLs above 25 wt%. The PVPVA release also slightly decreases with increasing DLs but remains at a much higher level. This means that with increasing DL, more and more RIT remains in the ASD, while PVPVA preferentially goes into solution. Thus, the release of RIT and PVPVA becomes non-congruent above 20 wt% DL. These results qualitatively agree with data from Indulkar et al. [2], who determined 25 wt% DL as the limit of congruency. The deviation in the congruency limit reported by Indulkar et al. [2] and the one observed in this work can be attributed to differences in experimental setups and methods, such as using compressed tablets instead of spray-dried particles and different mass ratios of ASD to dissolution medium (1 × 10^−3^ [2] vs. 1.43 × 10^−4^ in this work). Moreover, experiments of Indulkar et al. [2] were conducted at 37 °C. Since the RIT/water system (Appendix A) shows upper critical solution temperature (UCST) behavior, the miscibility gap at 37 °C is smaller compared to 25 °C.

#### 4.2.2. Release Mechanism of RIT/PVPVA ASDs

According to Figure 1 and Figure 2, both amorphous phase separation (in the ASD at low water concentrations) and liquid–liquid phase separation (in the dissolution medium at high water concentrations) occur when an ASD composed of amorphous RIT and PVPVA is exposed to water. (In equilibrium, the water activity in the ASD equals the water activity in the surrounding phase, be it a liquid or a vapor. For the latter, the water activity in the liquid equals RH.As shown in Figure 2, all RIT/PVPVA ASDs considered in this work at high RHs (97% RH mimics conditions during dissolution the most) absorb water to the extent that two coexisting amorphous phases evolve within the dissolving ASD. As noted in Table 2, RIT and PVPVA have glass-transition temperatures of 325.99 K and 380.74 K, respectively. Thus, ASDs with low RIT content (high PVPVA mass fractions) have lower molecular mobilities and, consequently, higher viscosities than ASDs with high RIT content, as also shown in the literature for other ASDs [35]. Therefore, the kinetics of amorphous phase separation in ASDs with low RIT content is much slower than in ASD with high RIT DLs, as also shown by Shi et al. via X-ray fluorescence imaging [8].

Thus, the ASD with 10 wt% RIT most likely completely dissolves before amorphous phase separation occurs within the ASD. The latter, therefore, does not affect the release behavior, and consequently, complete and simultaneous release of both RIT and PVPVA was observed as congruent release profiles over time (Figure 5). Moreover, due to the low RIT content, the release rate of PVPVA from the 10 wt% ASD is very similar to the one of pure PVPVA. This release mechanism is referred to as carrier-controlled or polymer-controlled.

With increasing RIT content, the kinetics of amorphous phase separation increase [8], whereas, at the same time, the dissolution kinetics decrease due to the lower polymer content in the ASD. This means that phase-separation kinetics within the ASD become faster than dissolution kinetics, resulting in the formation of a PVPVA-rich phase and an RIT-rich phase prior to dissolution into the medium. Caused by dissimilarity in composition, the driving forces for dissolution of the PVPVA-rich domains, on the one hand, and of the RIT-rich phase, on the other hand, are very different. Thus, they release with quite different kinetics (non-congruent release). This effect becomes more and more pronounced for increasing DLs due to (1) faster amorphous phase separation, (2) slower dissolution kinetics, and (3) an increasing amount of RIT-rich phase (shown in Figure 4). This leads to the non-congruent release profiles observed for the ASDs with DLs between 20 wt% and 40 wt% (Figure 5).

Figure 6 shows the plot of the experimentally determined compositions of the dissolution medium during the course of the dissolution experiments for 10−40 wt% ASDs in the ternary phase diagram of the RIT/PVPVA/water system. For that purpose, the water mass fractions were determined based on the mass fractions of RIT and PVPVA measured over time (∑$w_{i}$(t) = 1) and shown in Figure 5. At the beginning of the dissolution experiments (t = 0), neither RIT nor PVPVA were dissolved. Therefore, the dissolution course starts at pure water (top of the ternary phase diagram) and then proceeds downward as overall concentrations of RIT and PVPVA increase over time.

It becomes obvious that the release profiles for all DLs enter the miscibility gap predicted by PC-SAFT (Figure 6), triggering liquid–liquid phase separation in the dissolution medium and leading to the formation of a second phase already observed as nanodroplets by Indulkar et al. [2] between 10 wt% and 30 wt% DL. The kink in the release curves indicates the occurrence of the second phase in the dissolution medium once the solubility of amorphous RIT is reached. Particularly from the longer-lasting experiments at high DLs, it becomes obvious that the kink exactly occurs at the PC-SAFT predicted boundary of the liquid–liquid miscibility gap. This is another impressive proof of the perfect match between PC-SAFT predictions and experimental validation.

Based on the PC-SAFT predicted phase diagram, as well as on the experimental observations in literature and in this work, it is thus safe to say that both amorphous phase separation in the ASD and liquid–liquid phase separation in the dissolution medium occurs for all DLs of RIT/PVPVA ASDs. For liquid–liquid phase separation to occur, dissolution of RIT and PVPVA into the dissolution medium needs to be completed to a certain degree, whereas for amorphous phase separation to occur, certain amounts of water need to penetrate the ASD. Water penetration, amorphous phase separation, ASD dissolution, and liquid–liquid phase separation depend on time. The interplay of the different kinetics of these processes then leads to the observed release effects.

## 5. Conclusions

This work provides thermodynamic insight into the release mechanism of RIT/PVPVA ASDs based on the ternary phase diagram of the RIT/PVPVA/water system. We are convinced that the results are also transferable to other slow-crystallizing APIs in polymer-based ASDs.

The ternary RIT/PVPVA/water phase diagram was predicted using PC-SAFT and found in excellent agreement with experimentally determined equilibrium compositions and mass phase ratios. To the best of our knowledge, this is the first time ever that a quantitative phase diagram for the system RIT/PVPVA/water is provided. Based on this phase diagram, amorphous phase separation in the ASD and liquid–liquid phase separation in the dissolution medium could be identified as similar phenomena explained by just one miscibility gap.

Exposing the RIT/PVPVA ASD to an aqueous dissolution medium leads to crucial, interdependent processes: water absorption by the ASD, triggering amorphous phase separation in the wet ASD as well as ASD dissolution with subsequent liquid-liquid phase separation in the dissolution medium. The interplay between the kinetics of these processes strongly depends on the DL in the ASD and determines the release behavior. For low DLs (i.e., high polymer content), amorphous phase separation is slow due to the high glass transition temperature of the ASD, causing low API molecular mobility. On the other hand, the ASD dissolution at low DLs is fast as the high amount of the hydrophilic polymer promotes the dissolution process. In combination, this means that the wet ASD dissolves before it is affected by amorphous phase separation leading to congruent release.

High DLs accelerate amorphous phase separation of the wet ASDs while simultaneously decelerating the dissolution kinetics. The higher the DL, the lower the glass transition temperature of the ASD, making it prone to faster amorphous phase separation within the wet ASD. Moreover, the dissolution rate drops due to the high DL of the hydrophobic RIT. While ASD dissolution is vanishingly slow, amorphous phase separation occurs within the ASD, forming a heterogeneous system. Dissolving the phase-separated ASD containing drug-rich, as well as polymer-rich, domains consequently leads to a non-congruent release of RIT and PVPVA.

During dissolution, being congruent or non-congruent, liquid–liquid phase separation in the dissolution medium might occur as soon as the solubility of amorphous RIT in the dissolution medium is exceeded. This phase separation is observed as nanodroplet formation, whereby the evolving droplets are rich in RIT. However, this does not affect the release profile any further.

## Figures and Tables

**Figure 1 pharmaceutics-14-01904-f001:**
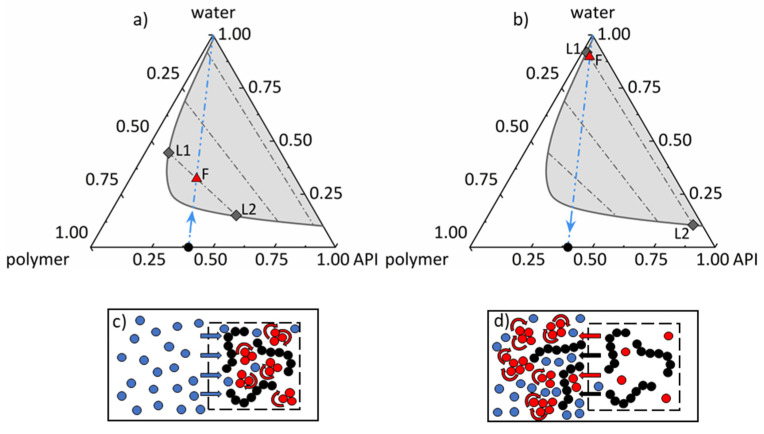
Schematic ternary phase diagram of an API/polymer/water system with schematic pathways of (**a**) water sorption and amorphous phase separation (according to (**c**)) and (**b**) ASD dissolution in water and liquid–liquid phase separation (according to (**d**)). The black circle depicts the dry ASD, and the blue dash-dotted line represents the path for (**a**) water sorption and (**b**) ASD dissolution. The gray solid line illustrates the miscibility gap (gray area) with gray dash-dotted tie lines. The red triangle denotes the feed point F for (**a**) amorphous phase separation and for (**b**) liquid–liquid phase separation. Gray diamonds denote the API-poor phase L1 and the API-rich phase L2.

**Figure 2 pharmaceutics-14-01904-f002:**
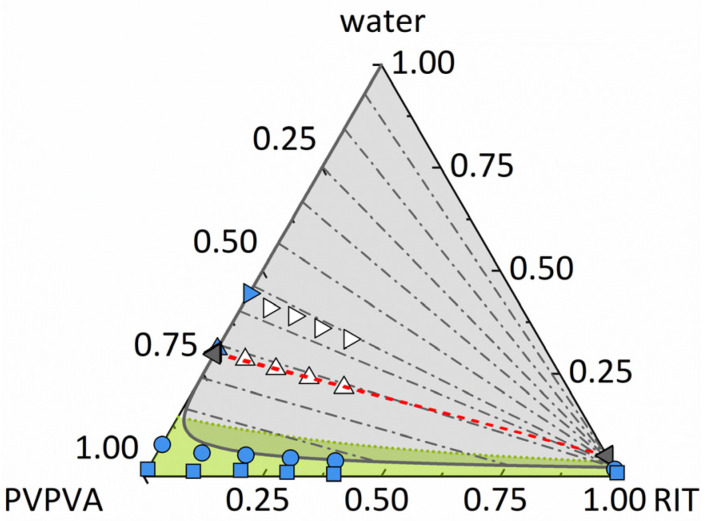
Ternary phase diagram of RIT/PVPVA/water at 25 °C and 0.1 MPa illustrating the PC-SAFT predicted miscibility gap (gray solid line enclosing the gray area), including tie lines (gray dash-dotted lines). Compositions are given in mass fractions. The green dotted line is the glass transition calculated using the Kwei equation. The glassy region is denoted as a green area. Symbols denote measured water-sorption data at 23% RH (squares), 53% RH (circles), 94% RH (triangles upwards), and 97% RH (triangles to the right) for the single-phase (blue symbols) and demixed systems (white symbols). Gray triangles represent the experimentally determined equilibrium compositions after amorphous phase separation at 94% RH, as summarized in Table 6.

**Figure 3 pharmaceutics-14-01904-f003:**
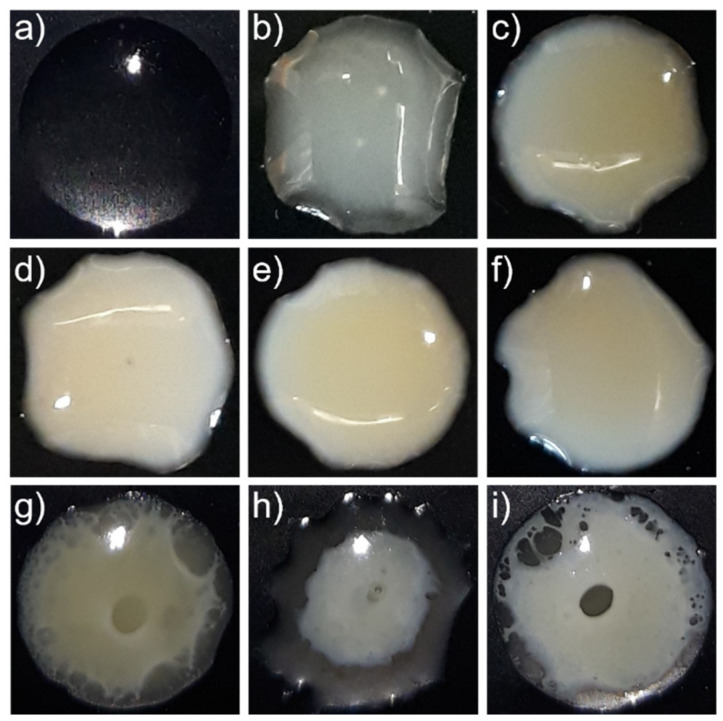
Photographs of ⌀8 mm RIT/PVPVA ASDs discs after two days of storage at 25 °C and 97% RH for DLs of (**a**) 0 wt%, (**b**) 5 wt%, (**c**) 10 wt%, (**d**) 15 wt%, (**e**) 20 wt%, (**f**) 25 wt%, (**g**) 30 wt%, (**h**) 35 wt%, and (**i**) 40 wt%.

**Figure 4 pharmaceutics-14-01904-f004:**
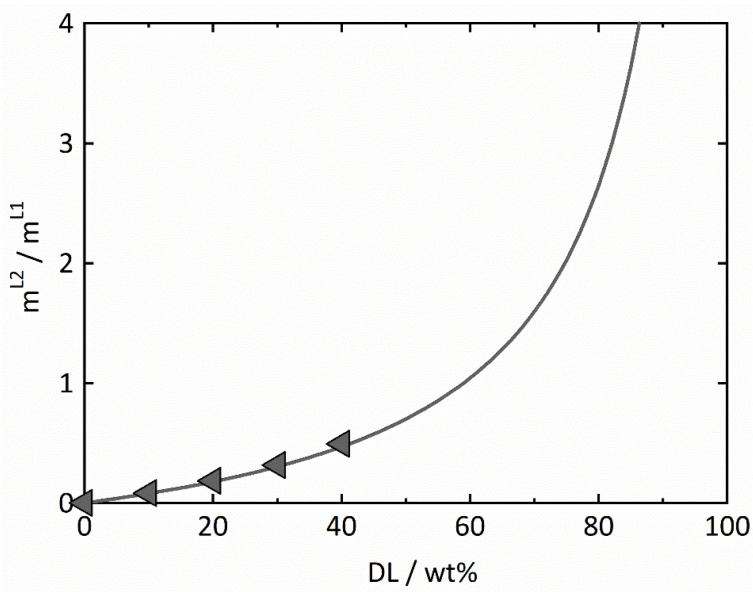
Equilibrium mass ratio of the RIT-rich (L2) and RIT-poor (L1) phases for RIT/PVPVA ASDs stored at 25 °C and 94% RH. The gray solid line denotes PC-SAFT-based prediction. Gray triangles are values obtained from mDSC experiments after optimizing Equation (14).

**Figure 5 pharmaceutics-14-01904-f005:**
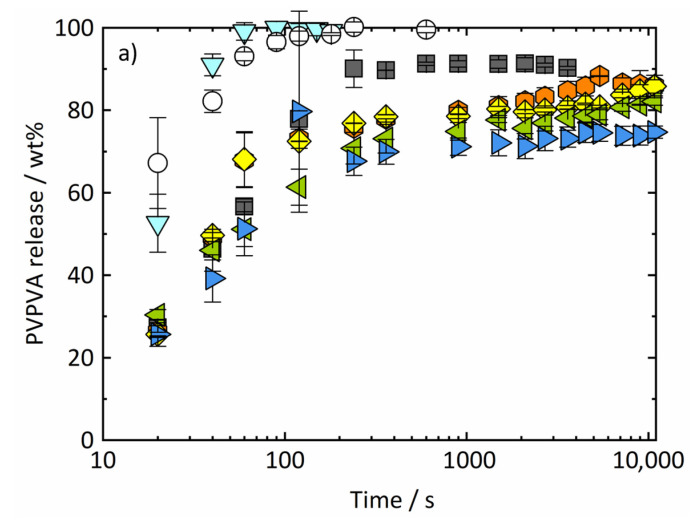
Release profiles of (**a**) PVPVA and (**b**) RIT in the dissolution medium over time at 25 °C and 0.1 MPa for ASDs with DLs of 0 wt% (light blue triangles downwards), 10 wt% (white circles), 20 wt% (gray squares), 25 wt% (orange hexagons), 30 wt% (yellow diamonds), 35 wt% (green triangles to the left), 40 wt% (dark blue triangles to the right), and 100 wt% (red triangles upwards) measured in this work.

**Figure 6 pharmaceutics-14-01904-f006:**
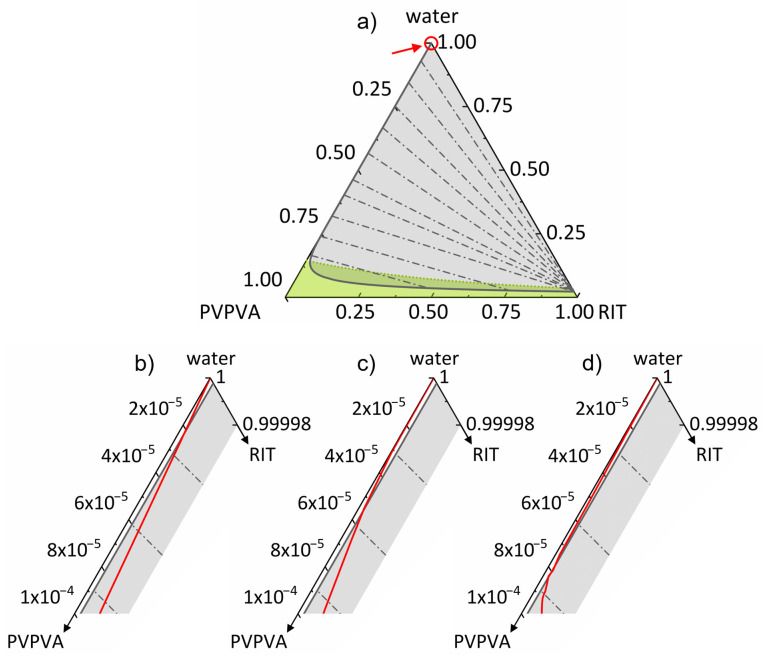
Zoom in the upper left section of the ternary phase diagram of RIT/PVPVA/water (lines in (**a**) are equal to Figure 2) at 25 °C and 0.1 MPa with the PC-SAFT calculated phase boundary (gray solid line left of the gray area) and tie lines (gray dash-dotted lines). The phase boundary indicates the solubility of amorphous RIT in the water/PVPVA solution. Compositions are given in mass fractions and correspond to the release data in Figure 5. Red solid lines are compositions of the dissolution medium during the course of the overall dissolution experiments of ASDs with DLs of (**b**) 10 wt% (t ≤ 20 s), (**c**) 20 wt% (t ≤ 120 s), (**d**) 25 wt% (t ≤ 3600 s), (**e**) 30 wt% (t ≤ 10,800 s), (**f**) 35 wt% (t ≤ 10,800 s), and (**g**) 40 wt% (t ≤ 10,800 s).

**Table 1 pharmaceutics-14-01904-t001:** Pure-component properties of RIT form II and of water at 0.1 MPa.

**Component ** i	$T_{i}^{SL}/K$	$\Delta h_{i}^{SL}/kJ\;mol^{-1}$	$\Delta c_{p,i}^{SL}/J\;mol^{-1}\;K^{-1}$
RIT form II [15]	398.12	63.20	224.16
water [16]	273.15	6.01	38.08

**Table 2 pharmaceutics-14-01904-t002:** Pure component densities and glass transition temperatures of RIT, PVPVA, and water.

Component i	$\rho_{i}/kg\;m^{-3}$	$T_{g,i}/K$
RIT	1150 *	325.99 ^†^
PVPVA	1190 [20]	380.74 ^†^
water	1000 [21]	138.00 [22]

* The density of amorphous RIT was assumed to be 8% smaller than the density of crystalline RIT (1250 kg/m^3^) based on the common heuristics for organic glasses [23]. ^†^ Measured in this work.

**Table 3 pharmaceutics-14-01904-t003:** Kwei interaction parameters fitted to glass transition temperatures of single-phase binary mixtures of RIT, PVPVA, and water measured in this work.

Mixture	$q_{ij}/K$	ARD/%
RIT/PVPVA	−18.83	0.56
RIT/water	−304.87	0.66
PVPVA/water	34.82	1.34

**Table 4 pharmaceutics-14-01904-t004:** PC-SAFT pure-component parameters of RIT, PVPVA, and water.

**Component** i	$M_{i}/g\;mol^{-1}$	$m_{i}^{seg}M_{i}^{-1}/mol\;g^{-1}$	$\sigma_{i}/Å$	$u_{i}k_{B}^{-1}/K$	$\varepsilon^{A_{i}B_{i}}k_{B}^{-1}/K$	$\kappa^{A_{i}B_{i}}$	$N_{i}^{assoc}$
RIT [15]	721	0.0220	3.900	305.787	1041.0	0.02	4/4
PVPVA [25]	65,000	0.0372	2.947	205.271	0	0.02	653/653
water [26]	18.015	0.0669	$\sigma_{\mathrm{water}}$ *	353.950	2425.7	0.0451	1/1

* $\sigma_{\mathrm{water}}=2.7927+10.11\;\exp\left(-0.01755\;T/K\right)-1.417\;\exp\left(-0.01146\;T/K\right)$.

**Table 5 pharmaceutics-14-01904-t005:** PC-SAFT interaction parameters for mixtures of RIT, PVPVA, and water.

Mixture	$k_{ij,a}/K^{-2}$	$k_{ij,m}/K^{-1}$	$k_{ij,b}$
RIT/PVPVA [15]	0	0	0.019
RIT/water [15]	0	0.00006	−0.059
PVPVA/water ^†^	−7.0045 × 10^−6^	0.0052677	−1.103909

^†^ Fitted to vapor–liquid equilibrium data from literature [29] as well as to liquid–liquid equilibrium data measured in this work.

**Table 6 pharmaceutics-14-01904-t006:** Glass-transition temperatures and compositions of RIT/PVPVA ASDs stored at 94% RH.

DL/wt%	$T_{g,exp}^{L1}/{}^{\circ}C$	$T_{g,exp}^{L2}/{}^{\circ}C$	$T_{g,cal}^{L1}/{}^{\circ}C$	$T_{g,cal}^{L2}/{}^{\circ}C$	$w_{RIT}^{L1}$	$w_{PVPVA}^{L1}$	$w_{water}^{L1}$	$w_{RIT}^{L2}$	$w_{PVPVA}^{L2}$	$w_{water}^{L2}$
10	−22.32	13.54	−27.87	13.50	0	0.70	0.30	0.95	0	0.05
20	−21.41	15.82	−27.08	15.62	0	0.70	0.30	0.95	0	0.05
30	−23.13	14.44	−26.78	14.21	0	0.70	0.30	0.95	0	0.05
40	−22.83	14.85	−26.30	14.49	0	0.70	0.30	0.95	0	0.05

## Data Availability

Data is contained within the Article or Appendix A.

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
