# Peer review of "Explaining the Release Mechanism of Ritonavir/PVPVA Amorphous Solid Dispersions"

_pharmaceutics, 2022, doi:10.3390/pharmaceutics14091904_

Round 1

Reviewer 1 Report

The work entitled ‘Explaining the release mechanism of ritonavir/PVPVA amorphous solid dispersion’ presents a detailed description for the release mechanism of ritonavir from a poly(vinylpyrrolidone-co-vinyl acetate) (PVPVA) amorphous solid dispersion formulation in water. The release mechanism presented is based on thermodynamic phase diagrams. The authors reported a successful correlation between experimentally acquired data and modelling phase equilibria and thermodynamical properties of the three components (ritonavir, PVPVA, water) and the corresponding mixtures using the Perturbed-Chain Statistical Associating Fluid Theory (PC-SAFT). Dissolution experiments of the amorphous solid dispersion with drug loading percentages varying in the range of 0-40% were performed. The results and the discussion are supported by adequate computed and experimental data and are also well and clearly presented as the paper is well written. This topic fits perfectly in the scope of the journal and will add value to the field in terms of understanding the release mechanism of an active pharmaceutical ingredient from an amorphous solid dispersion pharmaceutical formulation. I recommend publishing this manuscript in Pharmaceutics with two minor observations:

1) Throughout the manuscript one can find a large number of abbreviations and reading the manuscript can become a little difficult to follow the idea presented, so maybe a list of abbreviations, perhaps presented in the supplementary material file, could be really helpful for the reader.

2) Line 380 – reference not found

Author Response

The authors thank the reviewer for the comments and suggestions. We have included an abbreviation list in the revised the manuscript. Also, the missing reference on line 380 has been fixed in the revised manuscript.

Reviewer 2 Report

The authors have performed an interesting study of the use of PC-SAFT to be able to predict the dissolution profile of a ritonavir/PVPVA system.  What is particularly noteworthy is the fact that the prediction is based on independently measured parameters that were then confirmed experimentally.

This work clearly builds upon studies that have been performed by the Taylor lab at Purdue University, as is noted in the many citations to their work.  That being said, I am not aware how much work (if any) the Taylor lab has done to establish a predictive model such as PC-SAFT, and it is good to see the Sadowski lab extending their previous work in the field to these systems.  

Overall I find that the data supports the authors conclusions, and is consistent between the humidity data and the dissolution data.  My only real comment is that the authors have only studied one system using four relative humidity conditions and one set of dissolution experiments.  For this to truly be a predictive tool, more studies need to be performed on a variety of systems to truly show its ability to predict dissolution properties, but I feel that this study is both well done and quite useful.

Author Response

The authors thank and appreciate the comments and suggestions very much. The tool will be extended to cover variety of systems.